# PYRAMIDAL DENOISING DIFFUSION PROBABILISTIC MODELS

## ABSTRACT

Recently, diffusion model have demonstrated impressive image generation performances, and have been extensively studied in various computer vision tasks. Unfortunately, training and evaluating diffusion models consume a lot of time and computational resources. To address this problem to allow training with even a single GPU, here we present a novel pyramidal diffusion model that can generate high resolution images starting from much coarser resolution images using a *single* score function trained with a positional embedding. This enables a neural network to be much lighter and also enables time-efficient image generation without compromising its performances. Furthermore, we show that the proposed approach can be also efficiently used for multi-scale super-resolution problem using a single score function.

## 1 INTRODUCTION

Diffusion models produce high quality images via reverse diffusion processes and have achieved impressive performances in many computer vision tasks. Score-based generative models (Song et al., 2021b) produce images by solving a stochastic differential equation using a score function estimated by a neural network. Denoising diffusion probabilistic models (DDPMs) (Ho et al., 2020; Sohl-Dickstein et al., 2015) can be considered as discrete form of score-based generative models. Thanks to the state-of-art image generation performance, these diffusion models have been widely investigated for various applications.

For example, Rombach et al. (2021) trained a diffusion model on the latent space of a convolutional neural network (CNN)-based generative model, which enabled various of tasks. Diffusion-CLIP (Kim & Ye, 2021) leveraged contrastive language-image pretraining (CLIP) loss (Radford et al., 2021) and the denoising diffusion implicit model (DDIM) (Song et al., 2021a) for text-driven style transfer. ILVR (Choi et al., 2021) proposed conditional diffusion models using unconditionally trained score functions, and CCDF (Chung et al., 2021) developed its generalized frameworks and their acceleration techniques. Also, recently proposed models (Nichol et al., 2021; Ramesh et al., 2022) have achieved incredible performances on text-conditioned image generation and editing.

In spite of the amazing performance and flexible extensions, slow training and generation speed remains as a critical drawback. To resolve the problem, various approaches have been investigated. Rombach et al. (2021); Vahdat et al. (2021) trained a diffusion model in a low-dimensional representational space provided by pre-trained autoencoders. DDIM (Song et al., 2021a) proposed deterministic forward and reverse sampling schemes to accelerate the generation speed. Song & Ermon (2020) proposed a parameterization of covariance term to achieve better performance and faster sampling speed. Jolicoeur-Martineau et al. (2021) used adaptive step size without any tuning. PNDM (Liu et al., 2022) devised a pseudo numerical method by slightly changing classical numerical methods (Sauer, 2011) for speed enhancement. Salimans & Ho (2022) reduced the sampling time by progressively halving the diffusion step without losing the sample quality. Denoising diffusion GANs (Xiao et al., 2021) enabled large denoising steps through parameterizing the diffusion process by multimodal conditional GANs. For conditional diffusion, a short forward diffusion steps of corrupted input can reduce the number of reverse diffusion step in SDEdit (Meng et al., 2021) and RePaint (Lugmayr et al., 2022), whose theoretical justification was discovered in in CCDF (Chung et al., 2021) using the stochastic contraction theory.

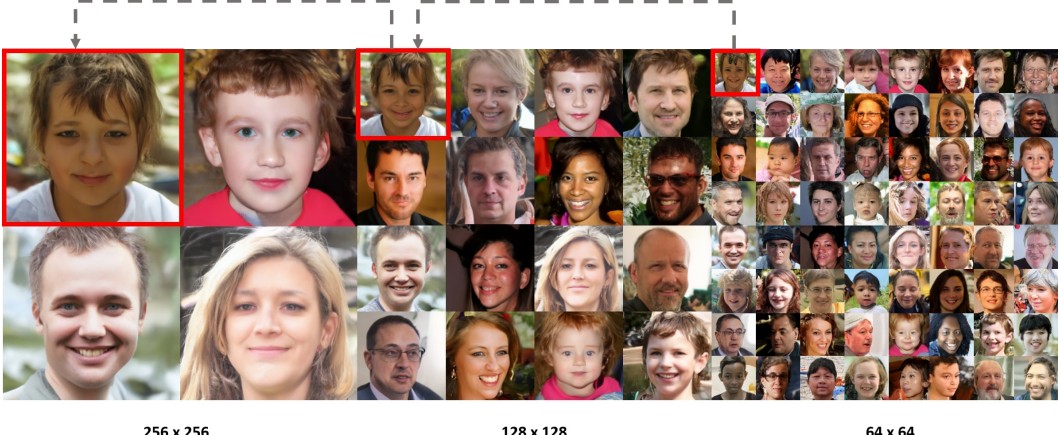

256 x 256                 128 x 128                 64 x 64

Figure 1: Progressive image generation from noises using the proposed method trained on FFHQ (Choi et al., 2020) dataset. Three different resolution images are generated from noise through reverse diffusion processes using a single model. In red boxes, the preservation of the semantic information at different resolution images is observed.

Alternatively, this paper addresses the slow sampling time issue in a similar manner to the method in Saharia et al. (2021) and Ho et al. (2022a) that refine low resolution images to high resolution using cascaded applications of multiple diffusion models. However, in contrast to (Saharia et al., 2021; Ho et al., 2022a), our model does not need to train multiple models, and can be implemented on a much lighter *single* architecture which results in speed enhancement in both training and inference without compromising the generation quality.

Specifically, in contrast to the existing diffusion models that adopt encoder-decoder architecture for the same dimensional input and output, here we propose a new conditional training method for the score function using positional information, which gives flexibility in the sampling process of reverse diffusion. Specifically, our pyramidal DDPM can generate a multiple resolution images using a single score function by utilizing positional information as a condition for training and inference. Fig. 1 shows the result of generated images in three different resolutions using only one model in the reverse diffusion process, which clearly demonstrates the flexibility of our method. Furthermore, as a byproduct, we also demonstrate multi-scale super-resolution using a single diffusion model.

The contribution of this work can be summarized as following:

- We propose a novel method of conditionally training diffusion model for multi-scale image generation by exploiting the positional embedding. In contrast to the existing diffusion model, in which the latent dimension and the output dimension are the same, in our method the output dimension can be arbitrarily large compared to the latent input dimension.

- Using a single score network, we mitigate high computation problem and slow speed issue of reverse diffusion process using a coarse-to-fine refinement while preserving the generation quality. The key element for this is again the positional encoding as a condition for the diffusion model.

- We present multi-scale super-resolution which recursively refines the image resolution using a single score model.

## 2 BACKGROUND

### 2.1 DENOISING DIFFUSION PROBABILISTIC MODELS

In DDPMs (Ho et al., 2020; Sohl-Dickstein et al., 2015), for a given data distribution $x_0 \sim q(x_0)$, we define a forward diffusion process $q(x_t|x_{t-1})$ as a Markov chain by gradually adding Gaussian

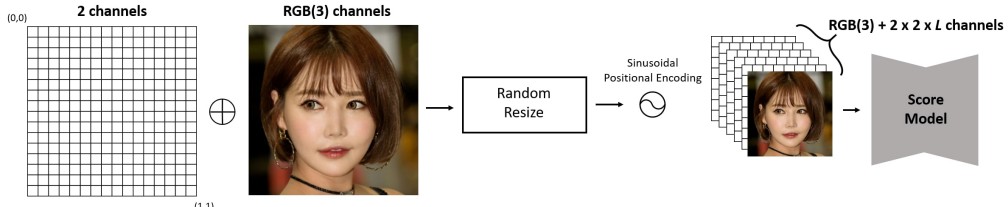

Figure 2: Our training scheme. Two dimensional coordinate information is concatenated with the input image and randomly resized to one of the target resolution. Then, two channels of coordinate values are encoded with the sine and cosine functions, and expanded to $2 \times 2 \times L$ channels where $L$ is the degree of positional encoding.

noise at every time steps $t$, where $\{\beta\}_{t=0}^{T}$ is a variance schedule:

$$q(\boldsymbol{x}_T|\boldsymbol{x}_0) := \prod_{t=1}^{T} q(\boldsymbol{x}_t|\boldsymbol{x}_{t-1}), \quad \text{where} \quad q(\boldsymbol{x}_t|\boldsymbol{x}_{t-1}) := \mathcal{N}(\boldsymbol{x}_t; \sqrt{1-\beta_t}\boldsymbol{x}_{t-1}, \beta_t \boldsymbol{I}). \quad (1)$$

With well scheduled $\{\beta\}_{t=0}^{T}$, the forward process converts a data distribution to an isotropic Gaussian distribution as $t \to T$. Using the notation $\alpha_t = 1 - \beta_t$ and $\bar{\alpha}_t := \prod_{s=1}^{t} \alpha_s$, we can sample from $q(\boldsymbol{x}_t|\boldsymbol{x}_0)$ in a closed form:

$$\boldsymbol{x}_t = \sqrt{\bar{\alpha}_t}\boldsymbol{x}_0 + \sqrt{1-\bar{\alpha}_t}\boldsymbol{z}, \quad \text{where} \quad \boldsymbol{z} \sim \mathcal{N}(\boldsymbol{0}, \boldsymbol{I}). \quad (2)$$

As the reverse of the forward step $q(\boldsymbol{x}_{t-1}|\boldsymbol{x}_t)$ is intractable, DDPM learns parameterized Gaussian transitions $p_\theta(\boldsymbol{x}_{t-1}|\boldsymbol{x}_t)$. The reverse process is defined as Markov chain with learned mean and fixed variance, starting from $p(\boldsymbol{x}_T) = \mathcal{N}(\boldsymbol{x}_T; \boldsymbol{0}, \boldsymbol{I})$:

$$p_\theta(\boldsymbol{x}_{0:T}) := p_\theta(\boldsymbol{x}_T) \prod_{t=1}^{T} p_\theta(\boldsymbol{x}_{t-1}|\boldsymbol{x}_t), \quad \text{where} \quad p_\theta(\boldsymbol{x}_{t-1}|\boldsymbol{x}_t) := \mathcal{N}(\boldsymbol{x}_{t-1}; \boldsymbol{\mu}_\theta(\boldsymbol{x}_t, t), \sigma_t^2 \boldsymbol{I}). \quad (3)$$

where

$$\boldsymbol{\mu}_\theta(\boldsymbol{x}_t, t) := \frac{1}{\sqrt{\alpha_t}}\Big(\boldsymbol{x}_t + (1-\alpha_t)\boldsymbol{s}_\theta(\boldsymbol{x}_t, t)\Big), \quad \boldsymbol{s}_\theta(\boldsymbol{x}_t, t) = -\frac{1}{\sqrt{1-\bar{\alpha}_t}}\boldsymbol{z}_\theta(\boldsymbol{x}_t, t) \quad (4)$$

Here, $\boldsymbol{s}_\theta(\boldsymbol{x}_t, t)$ is a score function and $\boldsymbol{z}_\theta(\boldsymbol{x}_t, t)$ is trained by optimizing the objective

$$\min_\theta L(\theta), \quad \text{where} \quad L(\theta) := \mathbb{E}_{t, \boldsymbol{x}_0, \boldsymbol{z}}\Big[\|\boldsymbol{z} - \boldsymbol{z}_\theta(\sqrt{\bar{\alpha}_t}\boldsymbol{x}_0 + \sqrt{1-\bar{\alpha}_t}\boldsymbol{z}, t)\|^2\Big]. \quad (5)$$

After the optimization, by plugging the learned score function into the generative (or reverse) diffusion process, one can simply sample from $p_\theta(\boldsymbol{x}_{t-1}|\boldsymbol{x}_t)$ by

$$\boldsymbol{x}_{t-1} = \frac{1}{\sqrt{\alpha_t}}\Big(\boldsymbol{x}_t + (1-\alpha_t)\boldsymbol{s}_\theta(\boldsymbol{x}_t, t)\Big) + \sigma_t \boldsymbol{z} \ . \quad (6)$$

## 2.2 POSITIONAL EMBEDDING

Positional embedding or encoding is widely used in many recent studies in order to give locational information to deep neural networks. For example, Devlin et al. (2018); Dosovitskiy et al. (2020) add position embedding to every entry of inputs in the form of trainable parameters. Unlike trainable embedding, Tancik et al. (2020) proposed a method applying sinusoidal positional encoding of the coordinate values for various tasks such as image regression, 3D shape regression, or MRI reconstruction. As the wave is continuous and periodic, low dimensional information can be expanded to a high dimensional space of different frequencies. In particular, the distance between periodically encoded vectors can be easily calculated by a simple dot product, so that the relative positional information of the data is provided without any additional effort.

The positional information is also useful for training neural network for image generation. Authors in (Anokhin et al., 2021) modified StyleGAN2 (Karras et al., 2020) into a pixel-wise image generation model, which outputs RGB values from the input of 2D coordinate of pixels on image. The

---

**Algorithm 1** Pyramidal reverse diffusion for image generation. It starts from initialization with Gaussian noise at the coarsest resolution, and ends at HR.

---

**Require:** $T_f, T_s, \triangle t_s, \{\alpha_t^f\}_{t=1}^{T_f}, \{\alpha_t^s\}_{t=1}^{T_s}, \{\sigma_t^f\}_{t=1}^{T_f}, \{\sigma_t^s\}_{t=1}^{T_s}$

1: $\boldsymbol{x}_{T_f}^{LR} \sim \mathcal{N}(\boldsymbol{0}, \sigma_{T_f}^f \boldsymbol{I})$             ▷ Gaussian sampling at the low resolution

2: **for** $t = T_f$ to 1 **do**             ▷ Full reverse diffusion

3:      $\boldsymbol{x}_{t-1}^{LR} \leftarrow \frac{1}{\sqrt{\alpha_t^f}}(\boldsymbol{x}_t^{LR} + (1 - \alpha_t^f)\boldsymbol{s}(\boldsymbol{x}_t^{LR}, t, pos(\boldsymbol{i}), pos(\boldsymbol{j}))) + \sigma_t^f \boldsymbol{z}$

4: **end for**

5: $\boldsymbol{x}_0 \leftarrow \boldsymbol{x}_0^{LR}$

6: **while** $\boldsymbol{x}_0$ is not $HR$ **do**

7:      $\boldsymbol{x}_0, \boldsymbol{i}, \boldsymbol{j} \leftarrow U^{\times 2}(\boldsymbol{x}_0), U^{\times 2}(\boldsymbol{i}), U^{\times 2}(\boldsymbol{j})$             ▷ Upsample image and coordinate value

8:      $\boldsymbol{x}_{T_s \triangle t_s} \leftarrow \sqrt{\bar{\alpha}_{T_s \triangle t_s}^s}\boldsymbol{x}_0 + \sqrt{1 - \bar{\alpha}_{T_s \triangle t_s}^s}\boldsymbol{z}$             ▷ Sample at $T_s \triangle t_s$

9:      **for** $t = T_s \triangle t_s$ to 1 **do**             ▷ Scaled reverse diffusion

10:          $\boldsymbol{x}_{t-1} \leftarrow \frac{1}{\sqrt{\alpha_t^s}}(\boldsymbol{x}_t + (1 - \alpha_t^s)\boldsymbol{s}(\boldsymbol{x}_t, t, pos(\boldsymbol{i}), pos(\boldsymbol{j}))) + \sigma_t^s \boldsymbol{z}$

11:      **end for**

12: **end while**

13: **return** $\boldsymbol{x}_0$

---

models of (Yu et al., 2022; Skorokhodov et al., 2021) generate continuous video frame by giving position encoded temporal information to the image generative models. The decoder in (Lin et al., 2019) generates full images despite the model being only trained with image patches and their center coordinate. Also, Lin et al. (2021); Ntavelis et al. (2022) train their models through patches and their coordinates, which allows the generative model to bring out arbitrary size of images.

## 3 PYRAMIDAL DENOISING DIFFUSION PROBABILISTIC MODELS

In this section, we provide a detailed explanation of our method, called the Pyramidal Denoising Diffusion Probabilistic Models (PDDPM).

### 3.1 MULTI-SCALE SCORE FUNCTION TRAINING

Training with multi-scale images is feasible in CNN-based models as they rely on the convolution calculation with spatially invariant filter kernels (Albawi et al., 2017). Leveraging this simple but strong characteristic of the architecture, our goal is to train diffusion model such that it can understand different scale of the input by giving coordinate information as a condition. Specifically, we concatenate an input image and coordinate values of each pixels $(i, j)$, while $i, j \in [0, 1]$ are normalized value of its $xy$ coordinate. Then, random resizing to the target resolution, 64/ 128/ 256 in our case, is applied on the merged input. The resized coordinate values are encoded with sinusoidal wave, expanded to high dimensional space, and act as conditions when training as shown in Fig. 2. Specifically, the positional encoding function is given by

$$pos(\boldsymbol{\gamma}) = \Big[ \sin(2^0\boldsymbol{\gamma}), \cos(2^0\boldsymbol{\gamma}), \sin(2^1\boldsymbol{\gamma}), \cos(2^1\boldsymbol{\gamma}) \cdots \sin(2^{L-1}\boldsymbol{\gamma}), \cos(2^{L-1}\boldsymbol{\gamma}) \Big] \quad (7)$$

$$\boldsymbol{\gamma} = \begin{bmatrix} \gamma_1 & \gamma_2 & \cdots & \gamma_n \end{bmatrix}^\mathsf{T}$$

where $\gamma_i \in [0, 1]$ and $L$ denotes the degree of positional encoding and $n$ is dimension of vector. By denoting $\boldsymbol{i} = \begin{bmatrix} i_1 & i_2 & \cdots & i_N \end{bmatrix}^\mathsf{T}, \boldsymbol{j} = \begin{bmatrix} j_1 & j_2 & \cdots & j_N \end{bmatrix}^\mathsf{T}$ as the collection of the normalized $x$ and $y$ coodinates from the $N$ pixels of $\boldsymbol{x}_t$, the training cost function in (5) can be converted as

$$L(\theta) := \mathbb{E}_{t, \boldsymbol{x}_0, \boldsymbol{z}} \Big[ \|\boldsymbol{z} - \boldsymbol{z}_\theta(\boldsymbol{x}_t, t, pos(\boldsymbol{i}), pos(\boldsymbol{j}))\|^2 \Big]. \quad (8)$$

Benefited from the UNet-like model structure (Ronneberger et al., 2015), the cost function Eq. (8) is invariant to all different resolutions so that the optimization can be performed with only a single network. This simple idea of scale-free training of the score network significantly improves the network's flexibility of sampling process which will be discussed later. Importantly, this can also alleviate the problem of slow training and low batch size problems especially when training with limited resources, the latter of which is significant for higher performance generative tasks.

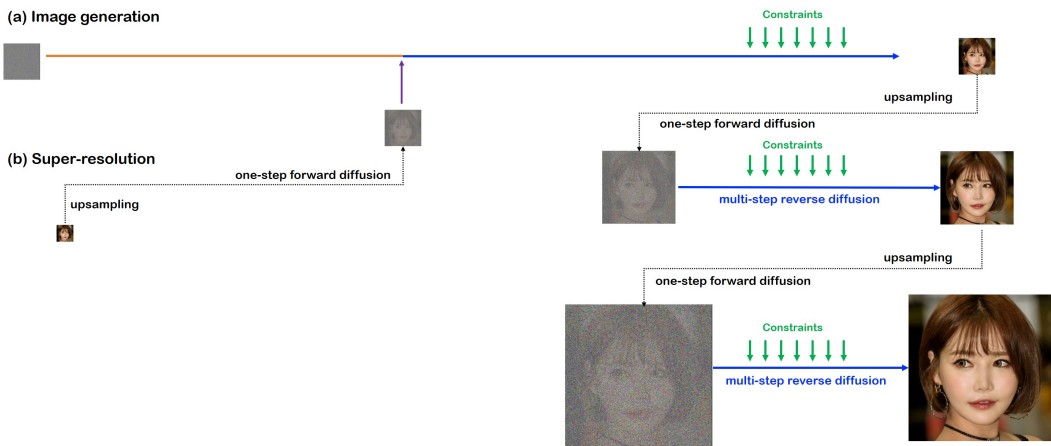

Figure 3: Proposed inference procedure for (a) image generation and (b) super-resolution. At the lowest resolution, full reverse diffusion is performed, which is then upscaled and forward diffused with additional noise. The CCDF (Chung et al., 2021) acceleration scheme is used as an acceleration scheme. For super-resolution, we imposes constraints in (10) at every step of the reverse process.

## 3.2 FAST REVERSE DIFFUSION THROUGH PYRAMIDAL REVERSE SAMPLING

Thanks to the multi-scale score function, the sampling speed, which is the most critical disadvantage of the diffusion models, can be also made much faster compared to a single full DDPM (Song et al., 2021a) reverse process. Although one may think that our method is similar to (Ho et al., 2022a; Jing et al., 2022) which trains additional score networks for sampling in lower dimensions, there are several important improvement in our method. First, our method only uses a *single* score model, which is trained with the positional encoding as explained in (5). Second, inspired by the CCDF acceleration scheme (Chung et al., 2021), the reverse sampling process can be further accelerated by using the lower-resolution reconstruction as an initialization for the next higher resolution reconstruction.

Specifically, as shown in Algorithm 1 and Fig. 3(a), we first set two different number of time steps, $T_f$ and $T_s$. Here, $T_f$ is a total time steps for full reverse diffusion process without any shortcut path which is applied on generating images of the lowest resolution; $T_s$ is a scaled time step for fast higher resolution reverse processes using non-Markovian diffusion process suggested in DDIM Song et al. (2021a). More specifically, starting from low resolution ($LR$) random Gaussian noise $\boldsymbol{x}_{t=T_f}^{LR} \sim \mathcal{N}(\boldsymbol{0}, \boldsymbol{I})$, the full reverse diffusion process is first performed, which is much faster compared to the reverse diffusion process at the maximum resolution. Then the generated $LR$ image and its position values are upscaled twice to produce the initialization for the next resolution. Then, the noises are added to the scaled image through forward diffusion with $T_s$, after which the reverse diffusion process is performed from $t = T_s \triangle t_s$ to $t = 0$ until the next higher resolution image is generated with $\triangle t_s \in (0, 1)$. This procedure is recursively applied to the next higher resolution images. Using this pyramidal image generation, the total sampling time can be significantly reduced compared to the single-resolution image generation at the highest resolution. Furthermore, according to the CCDF theory (Chung et al., 2021), the reverse diffusion process is a stochastic contraction mapping so that it reduces the estimation error from the forward diffused initialization at exponentially fast speed. Therefore, the required number of the reverse diffusion can be significantly reduced when a better initialization is used. This is why we use the previous resolution reconstruction as an initialization for the next finer resolution for further acceleration. For theoretical details, see (Chung et al., 2021).

## 3.3 PYRAMIDAL SUPER RESOLUTION WITH STABLE GRADIENT GUIDANCE

Recall that SR3 (Saharia et al., 2021) iteratively refines images from low to high resolution using two different score network modules. Likewise, Ho et al. (2022a); Ramesh et al. (2022); Song & Ermon (2020) generate low resolution images, and uses one or two different pretrained checkpoints for super-resolution. These diffusion models showed impressively high performance, but still re-

---

**Algorithm 2** Pyramidal reverse diffusion for super-resolution. It starts from initialization with input LR image $\bar{x}_0$ and ends at HR.

---

**Require:** $\bar{x}_0, T_s, \triangle t_s, \{\alpha_t^s\}_{t=1}^{T_s}, \{\sigma_t^s\}_{t=1}^{T_s}$

1: $x_0 \leftarrow \bar{x}_0$ ▷ Initialization using LR input image
2: **while** $x_0$ is not $HR$ **do**
3:      $x_0, i, j \leftarrow U^{\times 2}(x_0), U^{\times 2}(i), U^{\times 2}(j)$ ▷ Upsample image and coordinate value
4:      $x_{T_s \triangle t_s} \leftarrow \sqrt{\bar{\alpha}_{T_s \triangle t_s}^s} x_0 + \sqrt{1 - \bar{\alpha}_{T_s \triangle t_s}^s} z$ ▷ Sample at $T_s \triangle t_s$
5:      **for** $t = T_s \triangle t_s$ to 1 **do** ▷ Scaled reverse diffusion
6:          $x_{t-1} \leftarrow \frac{1}{\sqrt{\alpha_t^s}}(x_t + (1 - \alpha_t^s)s(x_t, t, pos(i), pos(j))) + \sigma_t^s z$
7:          $\hat{x}_0(x_t) := \frac{1}{\sqrt{\bar{\alpha}_t^s}}(x_t + (1 - \bar{\alpha}_t^s)s_\theta(x_t, t, pos(i), pos(j)))$
8:          $x_{t-1} \leftarrow x_{t-1} - \lambda \nabla_{x_t} \|D\hat{x}_0(x_t) - \bar{x}_0\|_2^2$
9:      **end for**
10: **end while**
11: **return** $x_0$

---

quires separately trained networks. However, in our case, only a single model with small number of forward and reverse processes is sufficient as explained in the following and Fig. 3(b).

Here, the reverse step can be guided towards the target by subtracting the gradient to the intended direction as suggested in (Ho et al., 2022b; Dhariwal & Nichol, 2021; Avrahami et al., 2022). Specifically, starting from (2), the denoised prediction of $x_0$ given $x_t$ is first computed by

$$\hat{x}_0(x_t) := \frac{1}{\sqrt{\bar{\alpha}_t}}(x_t + (1 - \bar{\alpha}_t)s_\theta(x_t, t, pos(i), pos(j))) \tag{9}$$

which is mixed with the reverse diffusion samples $x_{t-1}$ in (6) in terms of additional gradient to update the sample:

$$x_{t-1} \leftarrow x_{t-1} - \lambda \nabla_{x_t} \|D\hat{x}_0(x_t) - \bar{x}_0\|_2^2 \tag{10}$$

where $D$ denotes the down-sampling operator and $\bar{x}_0$ is the low-resolution measurement. Additionally, we use the CCDF acceleration scheme (Chung et al., 2021) similar to the aforementioned pyramidal image generation. One difference from the image generation is that even at the coarsest resolution, the full reverse sampling at the coarsest level is not necessary as the lower-resolution measurement can be utilized as an initialization after upsampling. See Algorithm 2 and Fig. 3(b).

## 4 EXPERIMENTS

### 4.1 DATASETS AND IMPLEMENTATION

We trained each score models for 1.2M iterations using the proposed method for FFHQ 256×256 (Choi et al., 2020) and LSUN-Church 256×256 (Yu et al., 2015) dataset, and proceeded 500k iteration for AFHQ-Dog 256×256 dataset. The model was trained with the batch size of 48/12/3 for 64/128/256 image resolution. FFHQ and LSUN-Church dataset were used to evaluate generation performances while AFHQ-Dog dataset was used for super-resolution. Our model is based on an improved version of DDPM [1]. To alleviate the memory and training speed problem in the limited resource environment using one GeForce 1080 Ti for all model training, we chose small size of model and the details are described in supplementary materials. Also we used 1000 diffusion steps for all training. We used Adam optimizer of 0.0001 learning rate. The degree of positional encoding $L$ in (7) was set to 6. For the base setting of the inference, we set $T_f = 1000$ and $T_s = 100$.

### 4.2 IMAGE GENERATION

We compared the quality of generated images among different reverse diffusion methods devised for fast sampling. Specifically, we trained a new model for numerical method (FON), DDIM, S-PNDM

---

[1]`https://github.com/openai/improved-diffusion`

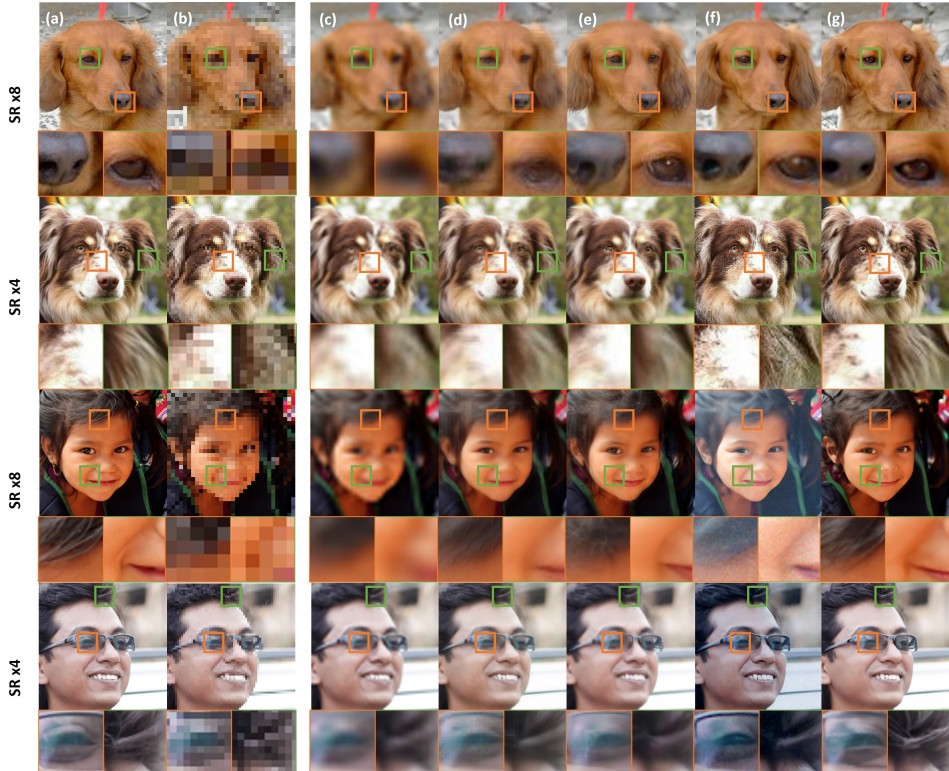

Figure 4: Result of super-resolution on FFHQ and AFHQ-dog dataset. Upper row shows the results of ×8 SR and bottom row is ×4 result. (a) Ground Truth, (b) low resolution images, the results by (c) cubic interpolation, (d) SRGAN, (e) ILVR, (f) MCG on AFHQ and SR3 on FFHQ, and (g) the proposed method.

Table 1: Frechet Inception distance (FID↓) comparison using 10k generated images. Sampling speed was also calculated by comparing with a full reverse diffusion process. Speed for the baselines were calculated using 100 sampling steps (gray-colored cells). The models are trained for 1.2M iterations for fair comparison.

| Dataset | Method | 10 | 20 | 50 | 100 | 200 | - | Params | Speed |
|---------|--------|-----|-----|-----|-----|-----|-----|--------|-------|
| **FFHQ** | FON | 26.71 | 18.91 | 16.70 | 16.13 | 16.22 | - | | ×9.13 |
| | DDIM | 37.87 | 27.71 | 20.45 | 18.30 | 16.85 | - | | ×10.1 |
| | S-PNDM | 28.59 | 22.62 | 18.13 | 16.71 | 16.48 | - | 114M | ×9.92 |
| | F-PNDM | 25.10 | 19.16 | 16.30 | 16.09 | 16.32 | - | | ×9.20 |
| | Ours | - | - | - | - | - | 15.78 | 16M | **×18.1** |
| **LSUN Church** | FON | 28.93 | 27.44 | 27.11 | 25.29 | 25.72 | - | | |
| | DDIM | 26.72 | 26.65 | 25.89 | 24.36 | 25.30 | - | | |
| | S-PNDM | 29.79 | 27.15 | 26.82 | 26.93 | 26.33 | - | 114M | - |
| | F-PNDM | 33.80 | 27.07 | 26.95 | 25.94 | 25.78 | - | | |
| | Ours | - | - | - | - | - | **14.07** | 16M | |

and F-PNDM (Liu et al., 2022) [2] for 1.2M iteration. The settings and training details are described in supplementary materials. As the model consists of 7 times larger parameters than ours, the training was performed on Quadro RTX 6000 to handle the low batch size problem. This means that the baselines are trained on better condition than ours. The model uses linear noise schedule and we chose 10, 20, 50, 100 and 200 steps for the total sampling steps. When sampling from ours, we set $\triangle t_s = 0.3$ so that all samplings procedure of higher resolution images can be done in only 30 steps. The sampling speed was calculated by comparing to the full diffusion step of the larger model.

---

[2] https://github.com/luping-liu/PNDM

Table 3: FID($\downarrow$), LPIPS($\downarrow$), PSNR($\uparrow$), SSIM($\uparrow$) score evaluation on super-resolution task. **Bold face**: best, underline: second best. (*:Unofficial re-implementation.)

| | FFHQ (256 × 256) | | | | | | | | AFHQ (256 × 256) | | | | | | | |
|---|---|---|---|---|---|---|---|---|---|---|---|---|---|---|---|---|
| SR facotr | ×4 | | | | ×8 | | | | ×4 | | | | ×8 | | | |
| Method | FID | LPIPS | PSNR | SSIM | FID | LPIPS | PSNR | SSIM | FID | LPIPS | PSNR | SSIM | FID | LPIPS | PSNR | SSIM |
| Bicubic | 125.7 | 0.278 | 28.84 | 0.851 | 151.3 | 0.446 | 24.73 | 0.703 | 35.34 | 0.280 | 29.10 | 0.816 | 69.47 | 0.429 | 25.34 | 0.671 |
| SRGAN | **46.85** | 0.204 | **29.45** | 0.857 | 71.69 | 0.296 | **26.20** | **0.753** | 22.61 | 0.247 | **29.48** | **0.826** | 45.35 | 0.329 | **25.91** | **0.688** |
| SR3* | 59.31 | 0.291 | 20.56 | 0.725 | 91.57 | 0.404 | 18.90 | 0.612 | 27.89 | 0.345 | 20.15 | 0.730 | 35.09 | 0.401 | 18.85 | 0.622 |
| ILVR | 54.73 | 0.224 | 29.15 | 0.851 | 71.47 | 0.295 | 24.92 | 0.712 | 26.61 | **0.234** | 28.76 | 0.800 | 37.39 | 0.326 | 25.10 | 0.650 |
| MCG | - | - | - | - | - | - | - | - | 22.10 | 0.246 | 27.34 | 0.782 | **32.96** | 0.323 | 24.78 | 0.652 |
| Ours | 49.06 | **0.192** | 28.70 | **0.860** | **66.80** | **0.289** | 24.83 | 0.725 | **21.62** | 0.237 | 27.74 | 0.796 | 33.14 | **0.311** | 25.38 | 0.649 |

We sampled 10k images from each method and evaluated visual quality using Frechet Inception distance based on `pytorch-fid`[3]. The result on Table 1 shows that our method produces superior results compared to the baselines despite the faster sampling speed with much smaller architecture. This implies that the model focuses on generating realistic images at lower resolution and adding fine details at higher resolution. Also the results on LSUN-Church dataset show that the training of the baselines was incomplete, whereas our method can produce qualitative results even with much limited resources.

We can further speed up the reverse process by re-spacing total diffusion steps of lower dimension images. We changed $T_f = T_s = 100$ and studied the effect of $\triangle t_s$ on image quality. By fixing total diffusion steps same for all resolution, we compared FID score on generated image when $\triangle t_s$ value changes from 0.1 to 0.5. The sampling speed computation was done similar as done in Table 1. Setting $T_f = 100$ and $\triangle t_s = 0.1$ increased the speed to be 82.4 times faster than original full diffusion process, while generating convincing images. Table 2 shows that large sampling steps on high resolution produce

Table 2: Self-Comparison results. $T_f$ is fixed to 100 which is shorter than the setting in Table 1. FID score and sampling speed are measured in the same way as in Table 1.

| $\triangle t_s$ | 0.1 | 0.2 | 0.3 | 0.4 | 0.5 |
|---|---|---|---|---|---|
| FFHQ | 23.91 | 22.42 | 22.11 | 21.91 | **21.87** |
| LSUN-Church | 27.39 | 18.10 | 15.38 | **14.57** | 14.63 |
| Speed | ×**82.4** | ×50.3 | ×36.1 | ×28.1 | ×22.9 |

higher FID scores, but the sampling speed decreases proportionally to $\triangle t_s$. Also, compared to the baselines in Table 1, our method produced comparable results to the baseline despite the faster sampling speed.

## 4.3 SUPER RESOLUTION

Experiment was performed on two SR factors: ×4, and ×8. We have tested on FFHQ, AFHQ-Dog 256×256 dataset with our models and other baselines: SRGAN, SR3, ILVR, MCG Chung et al. (2022) and traditional bicubic upsampling method for comparison. For SRGAN and SR3, there was no checkpoint for appropriate evaluation, so the models were trained on 32→256, 64→256 for each dataset. Additionally, the original work of SR3 (Saharia et al., 2021) cascaded two ×4 upsamplers for super-resolution, but in this case we reported on the result of super-resolution with one model for each SR factors. We made a comparison through FID↓, LPIPS↓, PSNR↑ and SSIM↑. Here, LPIPS [4] was calculated using the open source of the perceptual similarity of VGG (Simonyan & Zisserman, 2014). As shown in Table 3, our method produced the best result on FID and LPIPS for most of the cases, but not on PSNR and SSIM. Although SRGAN provides the best PSNR and SSIM values, it contains many image artifacts and the results of our method are more realistic as show in Fig. 4.

## 4.4 ABLATION STUDY

We performed an ablation study on pyramidal super-resolution and positional encoding. The ablation study on super-resolution was done by removing some steps of the resolution while up-sampling, in order to see the effect of pyramidal super-resolution. For positional encoding, we

---

[3] https://github.com/mseitzer/pytorch-fid.git
[4] https://github.com/richzhang/PerceptualSimilarity

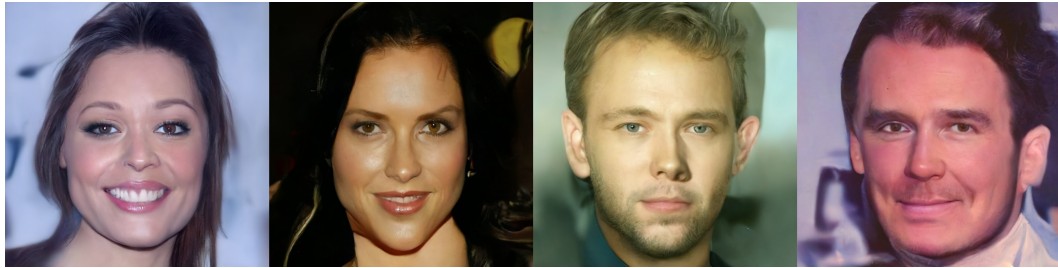

Figure 5: Generated images of at full resolution (1024×1024) by our method trained with only 256×256, 512×512 patches. The model had never seen full resolution image.

first trained a new model with multi-scale images without positional encoding. Second, we trained a score model with patches of $256 \times 256$ and $512 \times 512$ images for generation of $1024 \times 1024$ images.

Pyramidal super-resolution was evaluated by comparing the original method $32 \rightarrow 64 \rightarrow 128 \rightarrow 256$ with the up-sampling procedures of $32 \rightarrow 64 \rightarrow 256$, $32 \rightarrow 128 \rightarrow 256$ and $32 \rightarrow 256$. The evaluation was performed on both FFHQ and AFHQ-Dog dataset. The result in Table 4 shows that using all the resolution step of pyramidal super-resolution has achieved the best result. The CCDF (Chung et al., 2021) acceleration scheme of every resolution has shown quality improvement of reconstructed images.

Table 4: FID($\downarrow$), LPIPS($\downarrow$) scores of ablation study on super-resolution tasks.

| Method | FFHQ | | AFHQ | |
|---|---|---|---|---|
| | FID | LPIPS | FID | LPIPS |
| $32 \rightarrow 256$ | 72.78 | 0.308 | 45.99 | 0.340 |
| $32 \rightarrow 64 \rightarrow 256$ | 68.85 | 0.301 | 44.25 | 0.340 |
| $32 \rightarrow 128 \rightarrow 256$ | 68.33 | 0.291 | 36.80 | 0.328 |
| Original | **66.80** | **0.289** | **33.14** | **0.311** |

For the first ablation experiment on the positional encoding, we trained the model with the same experiment setting except for the positional encoding. It was trained for 800k iterations. Fig. 6 shows the result of the first experiment. Odd looking faces with multiple eyes, crushed face or dislocated facial features were generated. The result confirmed the importance of positional encoding.

Experiment on patch-wise learning for very high-resolution image generation was performed using one NVIDIA RTX 3090. The model was trained for 400k iterations with batch size of 6/2 on 256/512 scaled image patches of CelebA-HQ (Karras et al., 2017) dataset. When training, random resizing and cropping were used. Sampling was performed by setting $T_f = T_s = 100$. As seen in Fig. 5, although the score model had never seen a maximum resolution input, $1024 \times 1024$ images were generated through our fast pyramidal image generation.

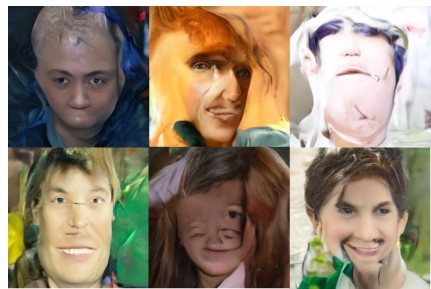

Figure 6: Images generated from a pyramidal DDPM without positional encoding. Facial features are not in proper location and some are crushed.

## 5 CONCLUSION

In this work, we proposed a novel Pyramidal DDPM which is trained with conditions on the positional information. We showed that this simple change improves the speed of reverse diffusion process and the performance of super-resolution and image generation. We also tested the effect of positional encoding by additional ablation experiments. Especially, without positional encoding, the model lost the ability to predict proper images at different resolution. Also patch-wise training further improved the flexibility of the score model, generating very high resolution images without using full resolution images. Given the significant advantages from a simple modification, we believe that our method may further mitigate high computation problems in the diffusion models and be used for many other applications.

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

# A    ADDITIONAL RESULTS

## A.1    IMAGE GENERATION

Visualization of generated samples from the comparison methods are provided in Fig. 7. We also provided additional samples of different resolution generated from a single score model on FFHQ and LSUN-Church dataset in Fig. 9 and Fig. 10. To verify the benefit of using positional encoding as condition, we added continuous magnification image generation ($\times 3.5$, $\times 5$) samples at Fig. 8. Especially, $320 \times 320 (\times 5)$ resolution results, which are larger than the maximum size of images used for training, shows the benefit of the proposed method.

## A.2    SUPER-RESOLUTION: INTERMEDIATE RESULTS

In this section, we provide intermediate results on pyramidal super-resolution task. Fig. 11 shows up-scaled images of each resolution starting from $32 \times 32$ on FFHQ and AFHQ-dog dataset.

## A.3    SUPER-RESOLUTION: SELF-COMPARISON RESULTS

We visualized the result in Table 4 in Fig. 12 and Fig. 13. Both of the result show that direct path from the low resolution image to the maximum resolution produce more details in the images. This implies the effect of iterative refinement method is not trivial.

# B    EXPERIMENTAL DETAILS

## B.1    COMPARISON METHODS

### B.1.1    IMAGE GENERATION

**Pyramdial DDPM**    We implemented Pyramdial DDPM based on the original code of improved diffusion model (Nichol & Dhariwal, 2021) [5]. We used the basic setting of the UNet model for generation of $64 \times 64$ images. Its stages use [64,128,256,512] channels from highest to lowest resolution with 1 residual block for each layer. We employ attention at $4 \times 4$ and $8 \times 8$ resolution to achieve better image generation quality on the lowest resolution. Exponential moving average over model parameters with a rate of 0.999 is used for evaluation. The model was trained using on GeForce 1080 Ti.

**FON, DDIM, PNDM**    We used the score function from PNDM (Liu et al., 2022) official github repository [6] as a baseline. Its stages use 2 residual blocks, [128,128,256,256,512,512] channels from highest to lowest resolution with 2 residual blocks for each layer and employed attention at $16 \times 16$. Exponential moving average over model parameters with a rate of 0.999 is used for evaluation. Adam optimizer with learning rate of 0.0002 was used. This setting is exactly the same as the original setting when training LSUN-Church dataset suggested by the author. Each model for LSUN-Church and FFHQ dataset was trained for 1.2M iterations for fair comparison with our model. FON, DDIM, S-PNDM and F-PNDM methods are used. The training was done with the batch size of 8 as the model was trained using Quadro RTX 6000.

### B.1.2    SUPER RESOLUTION

**SRGAN**    We trained a model from scratch on FFHQ and AFHQ-dog dataset using a code from SRGAN repository [7]. The model was trained for 500k iteration for each dataset. The model was trained with batch size of 8, 0.001 learning rate, 0.9 $\beta_1$ and 0.999 $\beta_2$ using Adam optimizer. We used one GeForce 1080 Ti for training.

---

[5] https://github.com/openai/improved-diffusion
[6] https://github.com/luping-liu/PNDM
[7] https://github.com/leftthomas/SRGAN

**SR3**   We used unofficial implementation for SR3 model from github repository[8]. The official model trained two separate model for $\times 4$ super-resolution, but we trained a single upsampler for $32 \to 256$ and $64 \to 256$. The training was done for 500k iteration with batch size of 2 for each dataset. The model consists of inner channels starting from 64 and 2 residual blocks. It was trained with 0.0001 learning rate using Adam optimizer. We used one GeForce 1080 Ti for training.

**ILVR, MCG**   We used pretrained network for both dataset from ILVR repository [9]. It was trained for 1M iterations for FFHQ, and 500k for AFHQ-dog. The checkpoint of AFHQ-dog is used for super-resolution task using MCG (Chung et al., 2022).

---

[8] `https://github.com/Janspiry/Image-Super-Resolution-via-Iterative-Refinement`
[9] `https://github.com/jychoi118/ilvr_adm`

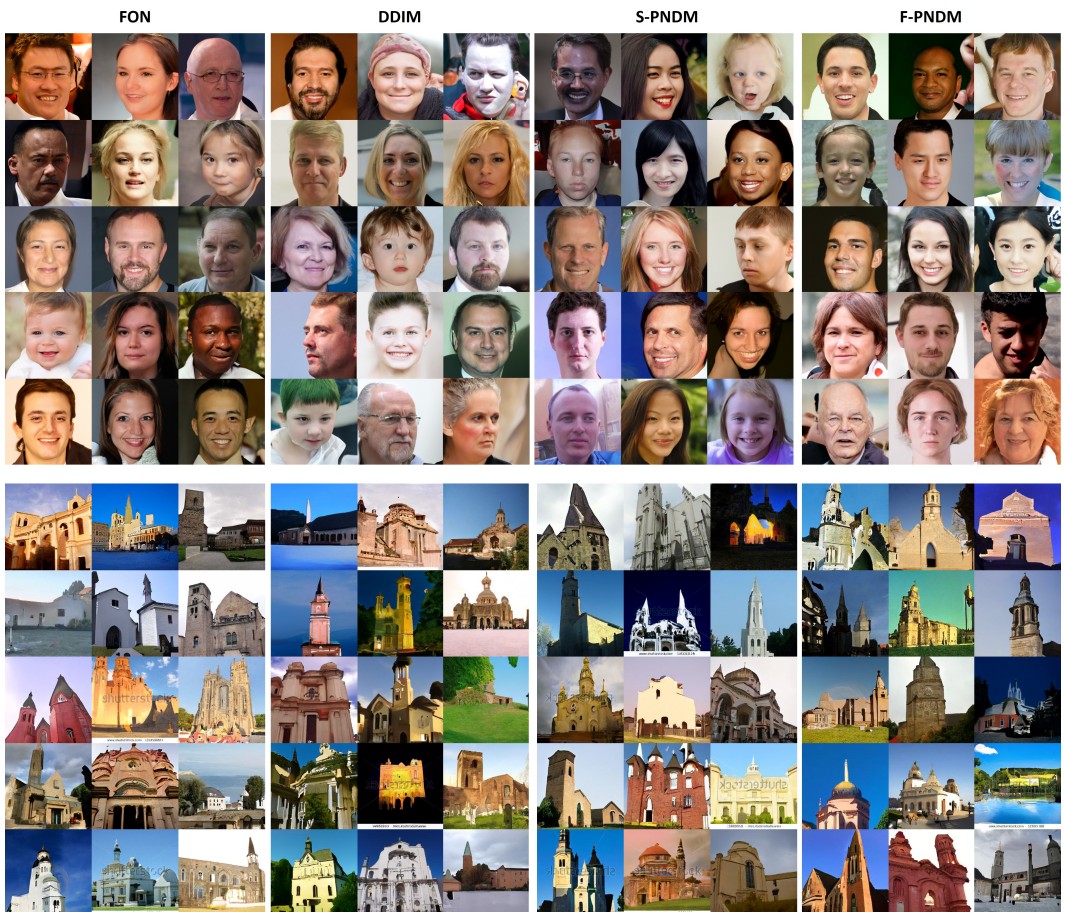

Figure 7: Generated images from comparison baselines(FON, DDIM, S-PNDM, F-PNDM) on FFHQ and LSUN-Church dataset using 100 reverse diffusion steps.

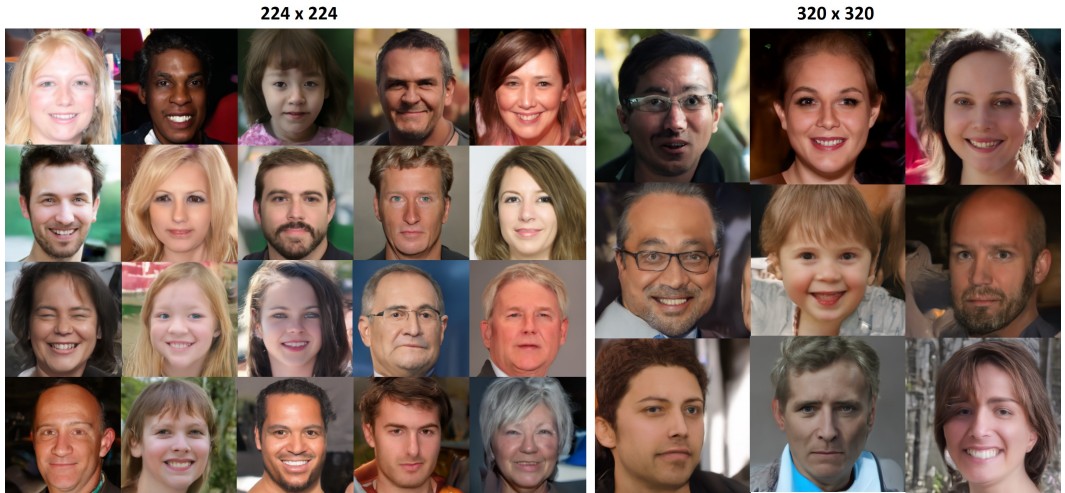

Figure 8: Continuous magnification image generation on $224 \times 224(\times 3.5)$ and $320 \times 320(\times 5)$ via Pyramidal DDPM.

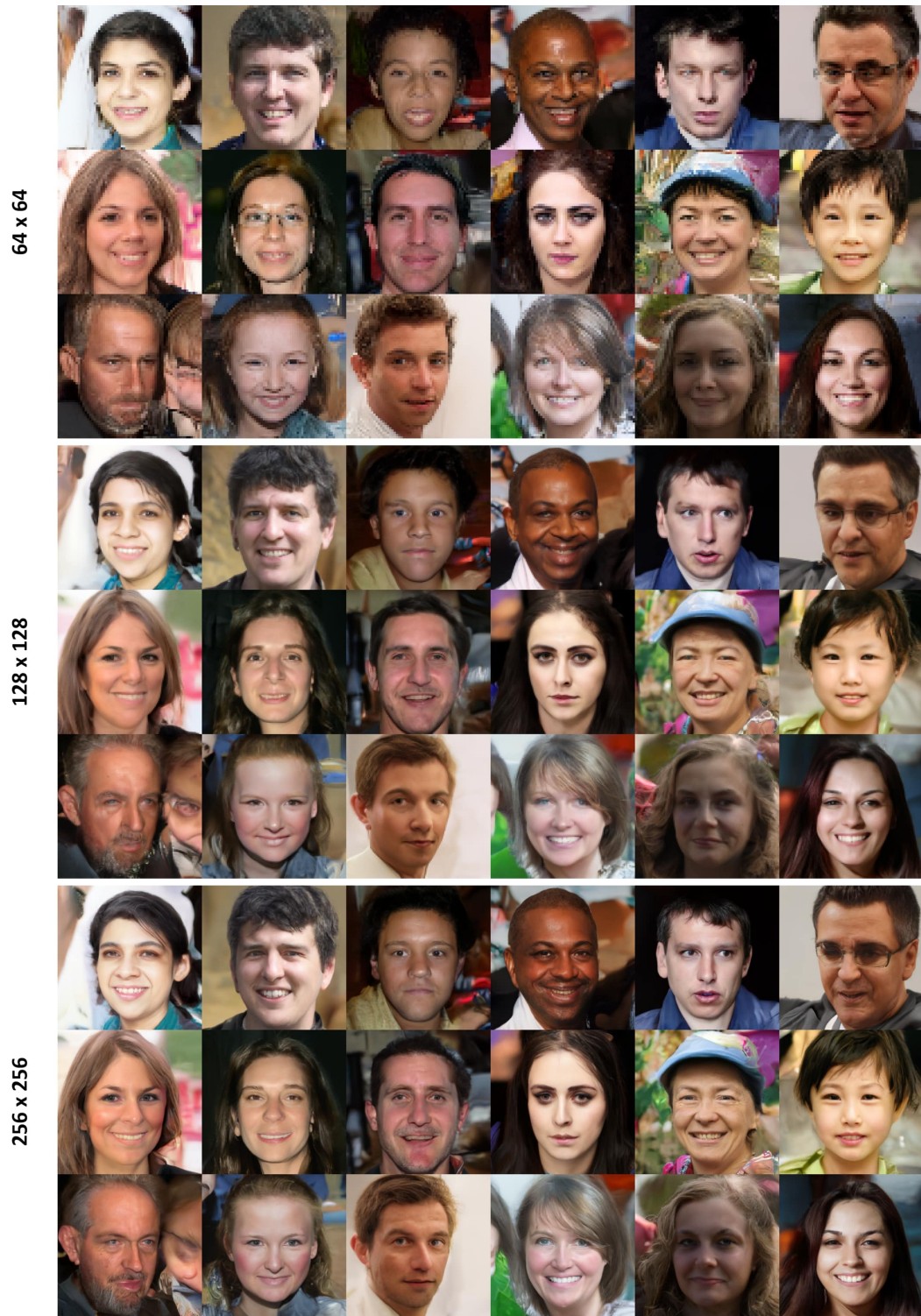

Figure 9: Additional generated images by our method trained from FFHQ dataset. Resolution increases starting from top to bottom row.

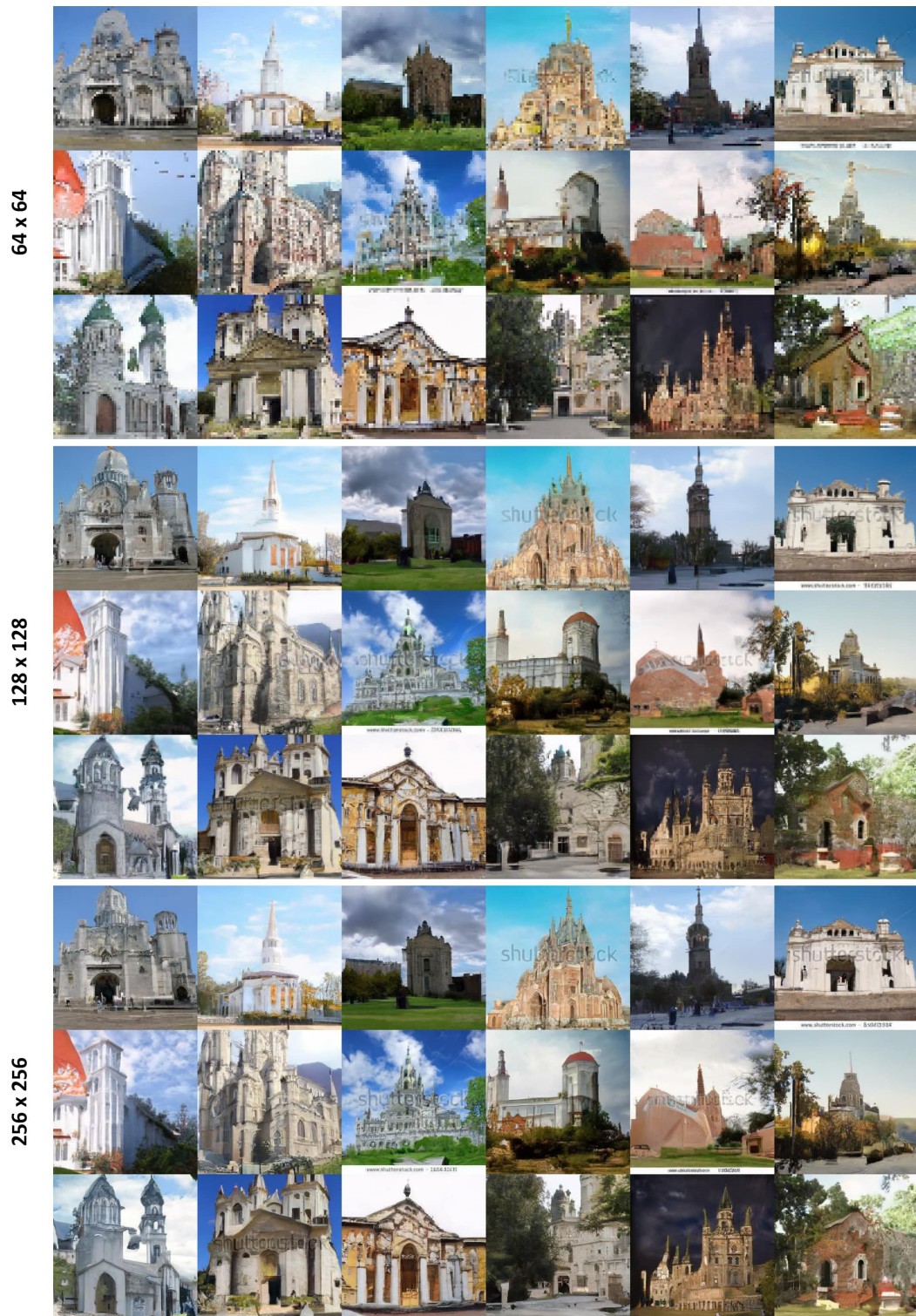

Figure 10: Additional generated images by our method trained from LSUN-Church dataset. Resolution increases starting from top to bottom row.

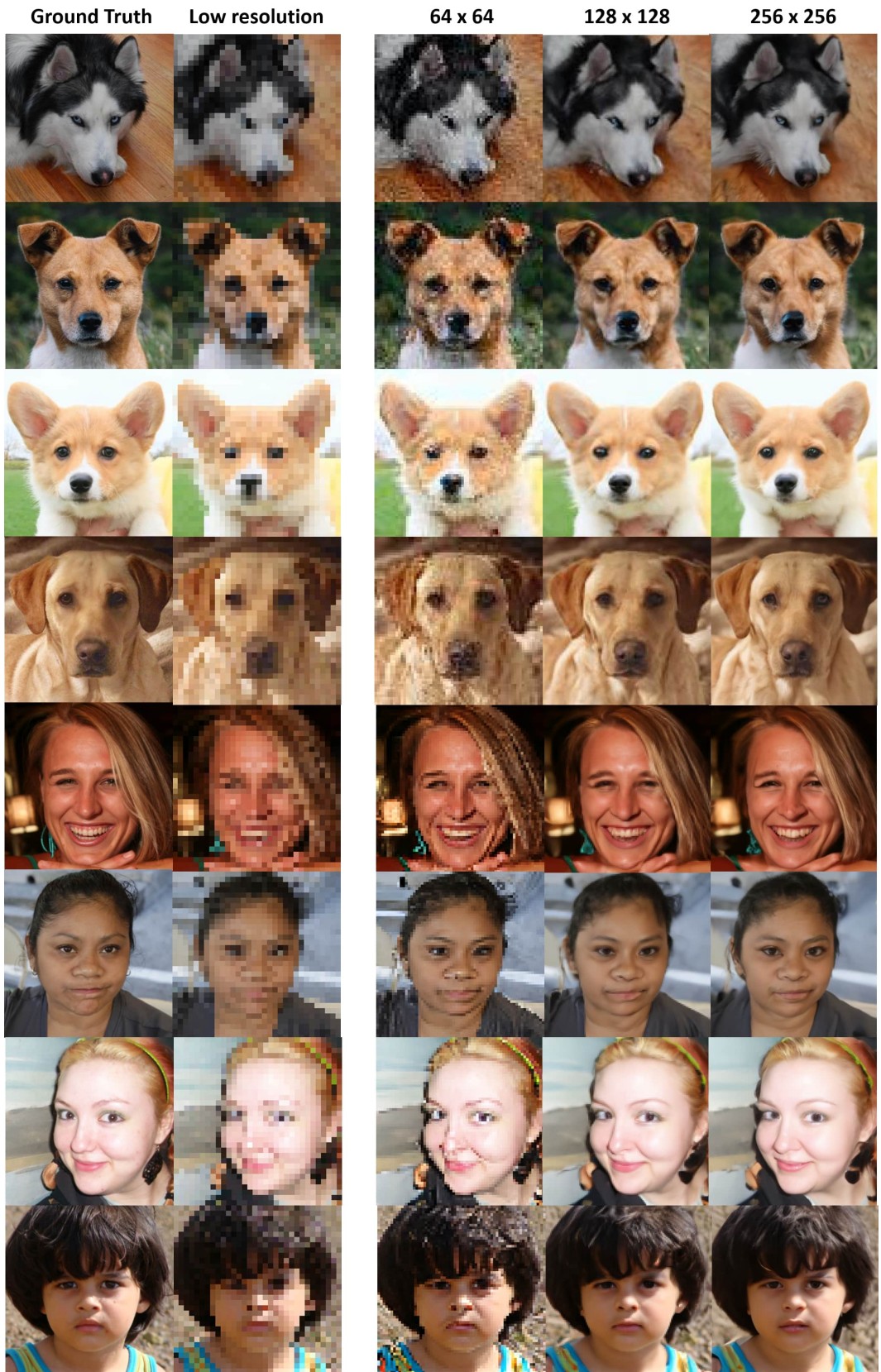

Figure 11: Intermediate results of pyramidal super-resolution on FFHQ and AFHQ-dog dataset.

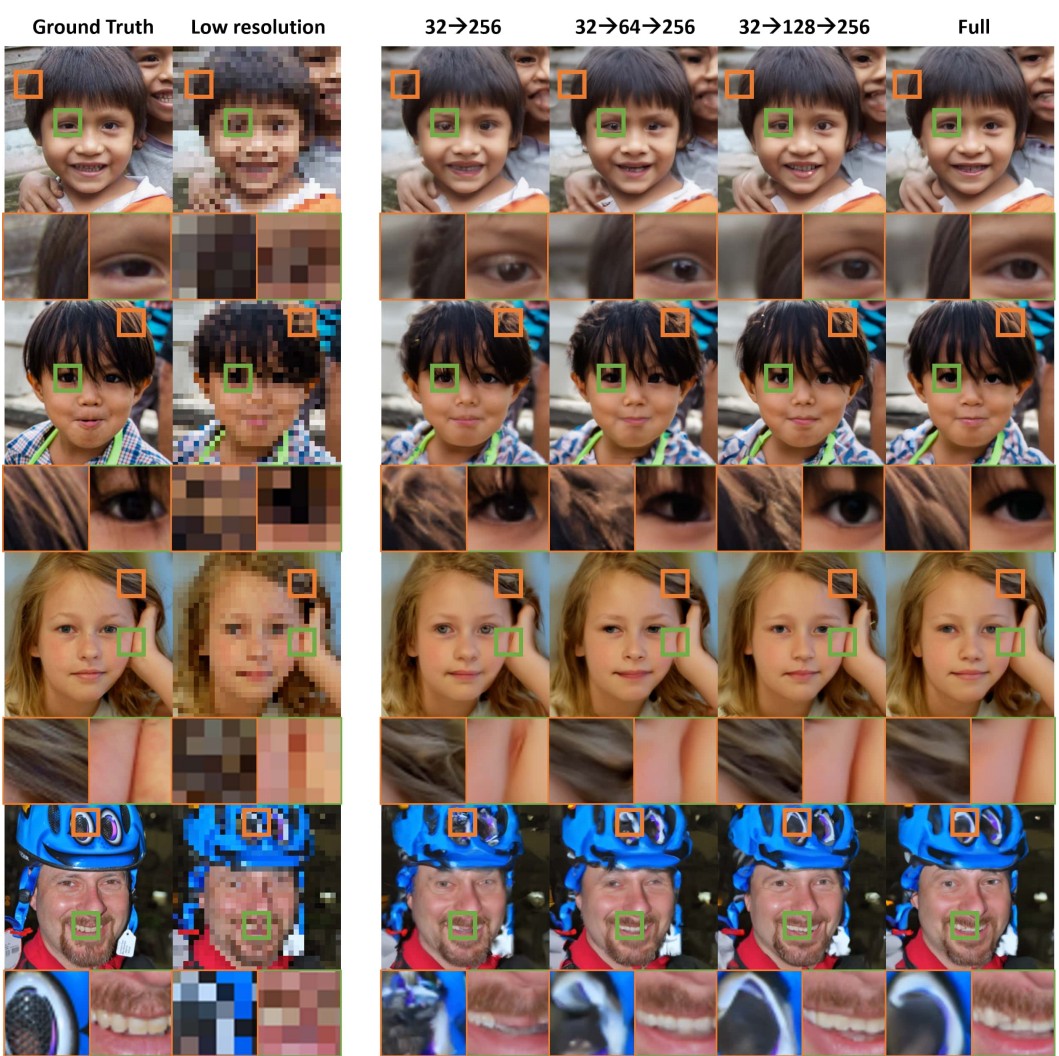

Figure 12: Visualization of the results of $\times 8$ super-resolution in Table 4 on FFHQ dataset.

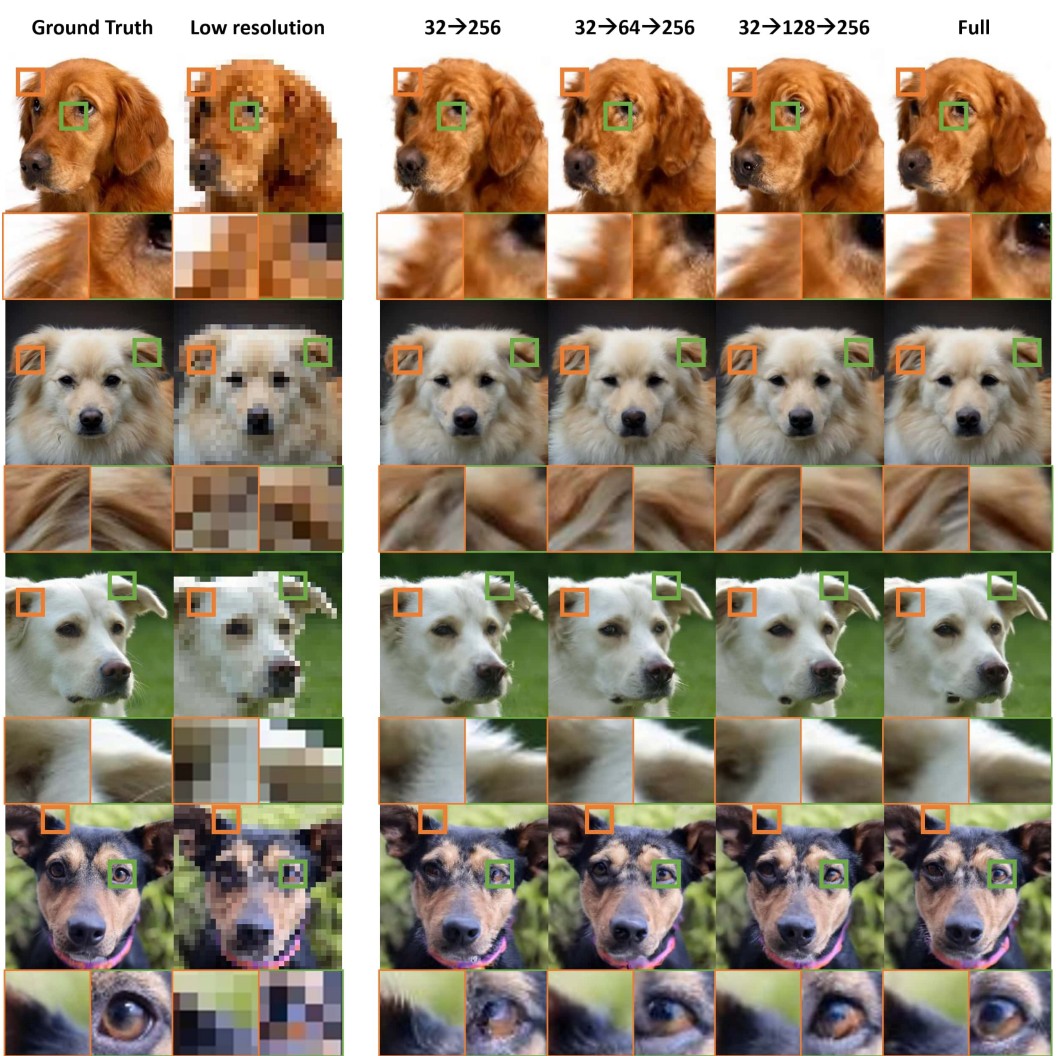

Figure 13: Visualization of the results of $\times 8$ super-resolution in Table 4 on AFHQ-dog dataset.

