# OpenReview forum: "Pyramidal Denoising Diffusion Probabilistic Models"
_ICLR.cc/2023/Conference — Submitted to ICLR 2023_

### Official Review · Reviewer_4NKS · 2022-10-25

**Confidence:** 4
**Clarity, Quality, Novelty And Reproducibility:** The description of the method is clear.
**Correctness:** 2
**Technical Novelty And Significance:** 3
**Empirical Novelty And Significance:** Not applicable
**Recommendation:** 3

**Details Of Ethics Concerns:**

The paper is about visual content generation and reconstruction, and specifically has examples of generating high-resolution facial images. It should at the very least discuss the potential for bias in these generated results.

**Strength And Weaknesses:**

# Strengths

- The idea of using positional embeddings to learn a single model to be applied across scales is interesting.

# Weaknesses

The main problem with the paper is in its evaluation.

- For the image generation task, the paper trains its own models for the competing methods, and the results are extremely surprising. While the competing methods are clustered together in their score, the proposed method does significantly better while also being faster and having an order of magnitude fewer parameters. For example, on the LSUN church dataset, the four other methods have scores ranging from 25.30 to 25.78, while the proposed method has a score of 14.07. The paper also doesn't include any visual examples of the images generated by the other methods, only its own.

Training requires setting and choosing many hyper-parameters, and given the significant gap in scores, it is possible that this difference is because of some hyper-parameter setting. Note that the other papers report their own numbers on different benchmarks. Why not train the proposed model under those settings, and report numbers comparing to those methods that way?

- For the super-resolution experiments, the closest method to the proposed one is SR3 --- which also does iterative scale diffusion, except with separate networks for each scale. The authors report results with their own trained version of SR3, and it appears to perform significantly worse. Note that the reported results show SR3 with a 7-9db worse PSRN than SRGAN while in SR3's own evaluation, it performed equivalently / slightly better than SRGAN (also on face images). Indeed, SR3 has 6-8db worse PSNR than bicubic interpolation! Moreover, visually inspecting the results, there seems to be a very visible change in the color temperature of the SR3 outputs from those of all other methods. This again suggests that there might be some issue with the implementation of SR3 used for testing.

In this case again, the SR3 paper reports results on a benchmark with a clear description of the training and evaluation sets. Why not replicate those settings and compare to SR3 on those reported numbers?

- In addition to the problematic comparisons to the baseline, the ablation studies could be improved. In particular, the ablation study should evaluate its specific novel contributions over previous methods. For example, it could show results from training a separate network for each scale, and demonstrate that with the proposed positional encoding, a single network comes close to matching results from such separate networks. It could show the benefit of CCDF and the projection steps, showing results when those were not applied.

- Given that the paper focuses on image generation, and in particular evaluates on face images, it should discuss the potential for bias in generated images.

**Summary Of The Paper:**

The paper proposes a new method to progressively generate or super-resolve to high-resolution images. It uses a diffusion approach for every up-sampling step, an idea it shares with Saharia et al. (SR3) and Ho et al. Its main innovation is in using a single model for all up-sampling steps (while SR3 used a separate model for each step).

**Summary Of The Review:**

Given the weaknesses in the evaluation as mentioned above, I do not think the paper is ready for publication. A new revision containing evaluations comparing to baselines in settings reported in those papers needs to undergo a fresh round of reviews. Moreover, the paper should include a discussion of bias and fairness concerns.

---

> ### Author Response · Authors · 2022-11-16
> **Response to Reviewer 4NKS**
>
> >**W1. The paper also doesn't include any visual examples of the images generated by the other methods, only its own.**
>
> Thanks for your feedback. We added the visual samples of comparison method in supplementary.
>
> >**W2. Training requires setting and choosing many hyper-parameters, and given the significant gap in scores, it is possible that this difference is because of some hyper-parameter setting. Note that the other papers report their own numbers on different benchmarks. Why not train the proposed model under those settings, and report numbers comparing to those methods that way?**
>
> We would like to assure the reviewer that the main purpose of the proposed method is to relieve the burden of training heavy networks. Using only a single GeForce 1080 Ti, we tried to compare our method and the baselines __when they are trained in the same condition.__ Especially, we empirically found out that the performance of the diffusion model highly depends on the batch size when training.
>
> At first, when training the baselines, we used the settings proposed in the original paper, but with the batch size of 2. However, the FID score was above 60 and the quality of the generated images was seriously bad. Instead, we used Quadro RTX 6000 which has twice the larger memory than ours, and trained the baselines using the batch size of 8. Despite the baselines were trained in a better condition that the proposed method, we could observe the superior results from our model.
>
> >**W3. For the super-resolution experiments, the closest method to the proposed one is SR3 which also does iterative scale diffusion, except with separate networks for each scale. The authors report results with their own trained version of SR3, and it appears to perform significantly worse. This suggests that there might be some issue with the implementation of SR3 used for testing. In this case again, the SR3 paper reports results on a benchmark with a clear description of the training and evaluation sets. Why not replicate those settings and compare to SR3 on those reported numbers?**
>
> We repeatedly tried to train new up-sampling diffusion models suggested in https://github.com/openai/improved-diffusion and https://github.com/Janspiry/Image-Super-Resolution-via-Iterative-Refinement. Again, we faced the same problem of image noise and hue deviation problems for both implementations using the same settings suggested in the original paper. Also, there are many similar cases which suffered from the same issues which can be found in the issues tab in the github repository. The only difference of the training was the batch size, so we guess there is a problem when training up-sampling diffusion model with low batch size (ours: 2, original: 128~256).
>
> To improve the clarity of the experiment, we have added super-resolution result on AFHQ dataset via MCG [1]. Figure and table in the paper are updated.
>
> > **W4. The ablation studies could be improved. In particular, the ablation study should evaluate its specific novel contributions over previous methods. For example, it could show results from training a separate network for each scale, and demonstrate that with the proposed positional encoding, a single network comes close to matching results from such separate networks. It could show the benefit of CCDF and the projection steps, showing results when those were not applied.**
>
> When CCDF acceleration is not used in image generation task, it will loss all information of lower resolution image as the reverse diffusion process has to start from the random noise. Therefore, our method would be worse without CCDF acceleration scheme.
>
> Instead, we have added additional experimental results on continuous magnification image generation, which improves the strength of the suggested method trained with positional encodings.
>
> > **W5. Given that the paper focuses on image generation, and in particular evaluates on face images, it should discuss the potential for bias in generated images.**
>
> As we used the images crawled from Flickr, the biases from the website are inherited. Also, our coarse-to-fine method sometimes changed the clothes, facial expression or sexuality of the lower-resolution image, which might have occurred by unintended correlation between some semantic features existed in the original FFHQ dataset. Lastly, the generative-model-based super-resolution method like ours may be affected by the bias when the input image is far apart from the original data distribution. For example, up-sampling a face image using the diffusion model trained with AFHQ dataset will not be possible.
>
> ___
> **References**
>
> [1] Chung et al, “Improving Diffusion Models for Inverse Problems using Manifold Constraints”, NeurIPS 2022

---

### Official Review · Reviewer_REZk · 2022-10-25

**Confidence:** 3
**Correctness:** 3
**Technical Novelty And Significance:** 3
**Empirical Novelty And Significance:** Not applicable
**Recommendation:** 6

**Clarity, Quality, Novelty And Reproducibility:**

This paper is clear and reproducible.

The introduction of spatial positional encoding for multi-scale cascaded generation with a single score function is novel and interesting.



**Strength And Weaknesses:**

Strength.
1. The idea of using a single score function to progressively generate higher-resolution images is interesting. And the use of spatial positional encoding seems an elegant solution for this task.

2. Introducing spatial positional encoding also enables patch-wise training of the diffusion model, which makes the training process much more flexible.

Weakness.
1. I do not fully agree with the claim of the super-resolution capability of the proposed method, although I do agree with the cascaded generation method. As seen from the red box of Fig.1, the 2x super-resolution breaks the consistency between the input lower-resolution image and the output image. (the mouth part). I think this is because every time a generated image is super-resolved, Gaussian noise is added during the forward diffusion process.  Although this is arguable because super-resolution can be ill-posed, however 2x super-resolution should not produce such a mismatch. I believe that's also why only on 4x~ super-resolution task, can this method outperforms existing works.

2. The comparison with SR3 seems weird. As seen in Figure.4, the generated images of SR3 have severe chromatic distortion. I think the unofficial implementation of SR3 may be buggy.



Other comments
Typos: 1. missing closing brackets in the reverse diffusion process in Algorithm.1 and Algorithm.2

**Summary Of The Paper:**

This paper is about training a multi-scale diffusion model in a unified framework. Specifically, it uses a casacaded mutli-scale generation process (from lower-resolution to higher-resolution). Unlike prior work which uses several score networks at each scale level. This paper uses a single network and proposes to use spatial positional encoding to make the network scale-aware and location aware. This spatial positional encoding also enables patch-wise training, which makes it feasible for large-scale image generation. Results demonstrated its ability to generate high-fidelity images at various scales.

**Summary Of The Review:**

I'm not fully aware of the field of diffusion models, so maybe I missed some prior works. From my point of view, this paper is novel and interesting. The contribution of this paper is clear. It uses spatial positional encoding to solve multi-scale cascaded generation with a single score function. Besides, this also enables patch-wise training.
The results also seem good.

---

> ### Author Response · Authors · 2022-11-16
> **Response to Reviewer REZk**
>
> Thanks for the feedback. To start with, we want to emphasize the main purpose of the proposed method. As we have only one GeForce 1080 Ti(12GB memory) GPU, we focused on solving the resource problem by relieving the burden of training heavy networks. __Instead of obtaining the better performance than the original works, we tried to improve the practical usage of Diffusion Models.__ To prove the efficiency of the proposed method, we decided to compare the network implemented in the original paper [1] and lighter network trained with positional encoding in the __“same training condition”__.
>
> > **W1. I do not fully agree with the claim of the super-resolution capability of the proposed method, although I do agree with the cascaded generation method. As seen from the red box of Fig.1, the 2x super-resolution breaks the consistency between the input lower-resolution image and the output image. (the mouth part).**
>
> Our method works differently on image generation and super-resolution. As shown in Figure. 3, we apply additional constraints in Eq. (10) for super-resolution in order to preserve the consistency of lower-resolution image. However, calculating Eq. (10) takes additional time, so it is unnecessary when generating a “new” image. The reviewer is kindly reminded that  the result in Fig. 1 only shows the result of image generation task, so that the exact consistency is not preserved. The preservation of the consistency between the input lower-resolution image and the output image for super-resolution task can be found in Fig. 4, 9, 10, 11.
>
> > **W2. The comparison with SR3 seems weird. As seen in Figure.4, the generated images of SR3 have severe chromatic distortion. I think the unofficial implementation of SR3 may be buggy.**
>
> We repeatedly tried to train new up-sampling diffusion models suggested in https://github.com/openai/improved-diffusion and https://github.com/Janspiry/Image-Super-Resolution-via-Iterative-Refinement. Again, we faced the same problem of image noise and hue deviation problems for both implementations. Also, there are many similar cases which suffered from the same issues which can be found in the issues tab in the github repository The only difference of the training was the batch size, so we guess there is a problem when training up-sampling diffusion model with low batch size (ours: 2, original: 128~256).
>
> For a fair comparison to improve the clarity of the experiment, we have added super-resolution result on AFHQ dataset via MCG [1]. Figure and table in the paper are updated.
>
> > **W3. Missing closing brackets in the reverse diffusion process in Algorithm.1 and Algorithm.2**
>
> Thanks, we corrected the typos.
> ___
> **References**
>
> [1] Chung et al, “Improving Diffusion Models for Inverse Problems using Manifold Constraints”, NeurIPS 2022

---

### Official Review · Reviewer_VnnC · 2022-10-26

**Confidence:** 4
**Correctness:** 2
**Technical Novelty And Significance:** 3
**Empirical Novelty And Significance:** 2
**Recommendation:** 5

**Clarity, Quality, Novelty And Reproducibility:**

Clarity/ Quality:  The paper is overall well-structured and easy to follow. However, some technical and experimental setting is not clearly stated. For example, the settings of hyperparameters $T_s/T_f$ and the use of CCDF  acceleration  are not clearly stated. Besides, the implementation details of Eq(10) are unclear. Finally, the introduction of  the network architecture is missing.

Novelty:  Using positional encoding for conditionally training diffusion models is an interesting direction, although based on existing work, it is overall a novel attempt for efficient and multi-scale image generation. However, I worry about the use of positional encoding is not well studied in this work with somehow insufficient numerical validations, which may diminish this proposal’s significance.

Reproducibility: Since this work is the integrations of existing works, the reproducibility seems not to be the issue.

**Strength And Weaknesses:**

Strengths:

I think the overall idea is interesting: designing a diffusion model that assembles the positional encoding as condition for fast sampling and multi-scale image generation. I believe this is an overall well-written paper with some new results that could spark further research in the interesting topic of image continuous scale generation. But I fear that the justification and benefit for the usage of positional encoding as condition input is not well justified through the experiments. Please find my technical comments below.

Weaknesses:

1, As cooperated by the authors, using positional information for training image generator networks is not new.

2, More importantly, for both image generation quality and SR restoration, the comparison with existing methods seems to be inaccurate. For image generation comparison, it is unclear why only the unofficial implementation results are reported. For example, in the original DDIM [Song.etal2021], the authors reported FID score of 10.84 on the LSUN-Church dataset when sampling 50 iterations.  Besides, the SR3 results show serious mean shift even for SRX4. The baseline methods seem to be re-implemented inappropriately. The pretrained DDPMs are public-available and reported similar results to the official implementation, such as https://github.com/openai/improved-diffusion.

3, Table 1. reports improved sampling speed of this work, comparing to baseline methods.  However, the reasons lead to such speedup is not clearly presented through ablation studies. As mentioned by the authors the CCDF (Chung et al., 2021) acceleration scheme is used as an acceleration scheme. Since CCDF is a general acceleration scheme, for fair comparison, all methods should use the same acceleration settings.

4, No continuous magnification results are showed in this paper. Since this work takes coordinate information as a condition and uses fully CNN architecture, it would be very informative to show the image generation or SR  results for continuous scale results (e.g., X3.2, X6.5).

5, Some implementation details are confusing. “We trained our model using FFHQ 256×256 (Choi et al., 2020), LSUN-Church 256×256 (Yu et al.,2015) datasets for 1.2M iterations”. Is this means that a single model is trained on multiple dataset ?


**Summary Of The Paper:**

This paper develops a variant (positional encoding) of conditional denoising diffusion probabilistic model (DDPM) with the goal of efficient sampling and multi-scale image generation while still maintaining the performance. By enforcing the gradient of the data-consistency during multi-stage sampling, the proposed diffusion model can also be used for image super-solution (SR). Both quantitative and qualitative results of this work are provided, comparing against several existing methods. The authors compared their method with several existing methods in terms of  sampling speed, image generation quality and SR performance.

**Summary Of The Review:**

Consequently, given the pros and cons on balance, I feel this is a very borderline paper, and I vote for borderline accept tentatively

---

> ### Author Response · Authors · 2022-11-16
> **Response to Reviewer VnnC**
>
> > **W1. Using positional information for training image generator networks is not new.**
>
> Even though using the positional information for training image generator network is not new, it is the first attempt to use the information on training the diffusion models.
>
> > **W2. More importantly, for both image generation quality and SR restoration, the comparison with existing methods seems to be inaccurate. For image generation comparison, it is unclear why only the unofficial implementation results are reported. Besides, the SR3 results show serious mean shift even for SRX4. The baseline methods seem to be re-implemented inappropriately.**
>
> - Image Generation:
>
> We would like to kindly remind the reviewer that the main purpose of the proposed method is to relieve the burden of training heavy networks. Using only a single GeForce 1080 Ti(12GB), we tried to compare our method and the baselines __when they are trained in the same condition.__ Especially, we empirically found out that the performance of the diffusion model highly depends on the batch size when training.
>
> For example,  when training the baselines, we used the settings proposed in the original paper, but with the batch size of 2. However, the FID score was above 60 and the quality of the generated images was seriously bad. Instead, we used Quadro RTX 6000(24GB) which has twice the larger memory than ours, and trained the baselines using the batch size of 8. Despite the baselines were trained in a better condition than the proposed method, we could observe the superior results from our model.
>
> - Super-resolution:
>
> We repeatedly tried to train new up-sampling diffusion models suggested in https://github.com/openai/improved-diffusion and https://github.com/Janspiry/Image-Super-Resolution-via-Iterative-Refinement. Again, we faced the same problem of __image noise and hue deviation problems__ for both implementations. Also, there are many similar cases which suffered from the same issues which can be found in the issues tab in the github repository. The only difference of the training was the batch size, so we guess there is a problem when training up-sampling diffusion model with low batch size (ours: 2, original: 128~256).
>
> For a fair comparison to improve the clarity of the experiment, we have added super-resolution result on AFHQ dataset via MCG [1]. Figure and table in the paper are updated.
>
> > **W3. Table 1. reports improved sampling speed of this work, comparing to baseline methods. However, the reasons lead to such speedup is not clearly presented through ablation studies. As mentioned by the authors the CCDF (Chung et al., 2021) acceleration scheme is used as an acceleration scheme. Since CCDF is a general acceleration scheme, for fair comparison, all methods should use the same acceleration settings.**
>
> CCDF proves  useful of full reverse diffusion process of conditional image generation when there is a better initial sample. Without having to start with the random noise, CCDF allow faster convergence of the image generation process. In the proposed method, a set of generated lower resolution images act as a good initial sample. The images are then up-sampled and noise is added to proceed higher resolution reverse diffusion process. Here, we can apply CCDF acceleration scheme.
>
> We would like to also kindly remind the reviewer that the other the baselines don’t have any initial states, so that CCDF scheme cannot be applied.
>
> > **W4. Continuous magnification ($\times{}3.2$, $\times{}6.5$ …etc) results will be informative.**
>
> We have tested continuous magnification image generation task on $\times{}3.5(64\rightarrow{}224)$ and $\times{}5(64\rightarrow{}320)$. We added the sampled images in supplementary. Even though the model was trained with the images with the maximum resolution of $256\times{}256$, the positional embeddings helped the model to generate larger size of the images.
>
> > **W5. Some implementation details are confusing. “We trained our model using FFHQ 256×256 (Choi et al., 2020), LSUN-Church 256×256 (Yu et al.,2015) datasets for 1.2M iterations”. Is this means that a single model is trained on multiple dataset?**
>
> We would like to remind the reviewer that the models are trained separately. We have clarified the implementation details.
> ___
> **References**
>
> [1] Chung et al, “Improving Diffusion Models for Inverse Problems using Manifold Constraints”, NeurIPS 2022

---

### Official Review · Reviewer_HWig · 2022-10-27

**Confidence:** 4
**Correctness:** 3
**Technical Novelty And Significance:** 3
**Empirical Novelty And Significance:** 2
**Recommendation:** 5

**Clarity, Quality, Novelty And Reproducibility:**

The clarity should be improved. The novelty and quality are okay. See more details in Strength And Weaknesses.

**Strength And Weaknesses:**

Strength:

- The core idea of incorporating positional encoding into the diffusion model is a valid contribution. It allows multi-scale sampling with a single score function model. The effectiveness of this strategy is verified with ablation study.

- The paper shows good sampling speed for diffusion-based image generation.

- The paper makes a good literature review about denoising diffusion models in the introduction, especially the works on applications of diffusion models.


Weakness:

- The results of the proposed algorithm do not look very good. While I understand this might be due to the small model size used in this paper, it will be important to show that the algorithm has the potential to achieve better performance by slightly enlarging model sizes.

- For the speed comparison, it is important to compare with existing methods that are also based on multi-scale sampling, for example, SR3 and Cascaded Diffusion Model.

- The paper is not well written, and some contents are hard to understand. For example:

(1) What is 2x2 in positional encoding in Figure 2?

(2) On Line 6 of Algorithm 1, I think it should be "while $x_0$ is 'not' HR do". Please clarify if I misunderstand this.

(3) On P6, the paper says "1000 diffusion steps for all training". It is not clear how these steps are split between different scales. Also, I don't know why the number of steps is different in inference: $T_f$=1000, $T_s$=100, which is more than 1000 in total.

(4) On P8, the paper mentioned Table 2(b) and Table 1(a) which however are never presented in the paper.

(5) On P9, the paper presents an experiment where the score model is trained with patches of 256×256 and 512×512 images for the generation of 1024×1024 images. What is this experiment for? How do we interpret this result?

(6) Typo: on P6, $T^f$ and $T^s$ should be $T_f$ and $T_s$; in Figure 3, diffsion --> diffusion

**Summary Of The Paper:**

This paper aims to solve the low-efficiency problem of diffusion models and presents a pyramidal diffusion model that can generate high-resolution images starting from much coarser resolution images using a single score function.
The key idea is to use a positional embedding, which enables a neural network to be much lighter and also enables time-efficient image generation without compromising its performance. The paper also shows that the proposed approach can be used for super-resolution using a single score function.

**Summary Of The Review:**

The usage of positional encoding in the diffusion model is interesting. However, the results are not quite convincing. Also, the paper has writing issues. See more details in Strength And Weaknesses.

---

> ### Author Response · Authors · 2022-11-16
> **Response to Reviewer HWig**
>
> > **W1. The results of the proposed algorithm do not look very good. It will be important to show that the algorithm has the potential to achieve better performance by slightly enlarging model sizes.**
>
> We would like to assure the reviewer that the main purpose of the proposed method is to relieve the burden of training heavy networks. Especially, we empirically found out the performance of the baseline models is critically deteriorated when trained with low batch size. For example, when we trained the baselines with the batch size of 2 (12GB GPU memory), the FID score exceeded 60, so we trained them with larger memory as mentioned in the paper. In contrary, the light network of ours alleviated the issue and achieved better performances. We could expect performance increase if we enlarge both the network and the batch size, but the suggested network better supports the main objective of the paper which aims to reduce the computational complexity of diffusion model training.
>
> > **W2. For the speed comparison, it is important to compare with existing methods that are also based on multi-scale sampling, for example, SR3 and Cascaded Diffusion Model.**
>
> As there is no official code for SR3 and Cascaded Diffusion Model, we couldn’t compare both FID and speed of the methods. Instead, we only measured speed of Cascaded Diffusion Model with the setting proposed in the original paper and compared with ours. __The proposed method spent 9.5 seconds and Cascaded Diffusion Model took 55.54 seconds for a single image generation.__ The main reason for the large speed gaps between them seems to be the size difference of the models. Furthermore, as our method starts from intermediate step via CCDF acceleration, the full reverse diffusion process is not required when up-sampling the generated low-resolution image, so the proposed method will be faster even when the size of the models are equivalent.
>
> > **W3. This paper is not well written, and some contents are hard to understand. For example :**
>
> **1. What is $2\times{}2$ in positional encoding in Figure 2?**
>
> The first 2 denotes $(i,j)$ coordinate of each pixel, and the next 2 refers to cosine and sine encodings for each $i$ and $j$. This means that, 2-dimensional coordinate vector of a single pixel is encoded into 4-dimensional vector for each frequency encoding. When $L$ is the degree of the positional encoding, then the total dimension of encoded vector becomes $2\times{}2\times{}L$.
>
> **2. On line 6 of Algorithm 1, ‘while’ should be ‘while is not’**
>
> Thanks for the careful reading. We have corrected it.
>
> **3. It’s hard to understand the number of $T_f =1000$ and $T_s = 100$ which is more than total diffusion steps $1000$.**
>
> Firstly, our Pyramidal DDPM model is trained with the total diffusion step of $1000$, which is one of the hyperparameters for the training. When sampling images from the model, we can respace the total diffusion steps via non-Markovian diffusion process suggested in DDIM [1], so we used $1000$ steps when sampling $64\times{}64$ images and respaced diffusion steps to $100$ when sampling higher resolution. We added the reference in Section 3.2 to clarify this.
>
> To give an example for image generation, we performed $T_f = 1000$ steps for $64\times{}64$ image generation, and performed one-step forward diffusion (noise adding) after bilinear up-sampling. As we set the amount of forward diffusion $\Delta{t}=0.3$ and $T_s = 100$, the reverse diffusion process can be done in only $30$ steps. As a result, total $1060$ steps are required for the image generation.
>
> **4. Table 1 (a) and Table 2(b) are never presented in the paper**
>
> We corrected the typos
>
> **5. What is the experiment of patch-wise training for?**
>
> We would like to kindly remind the reviewer that the main purpose of our research was to solve the heavy resource requirement problem for diffusion models. It is impossible to train a diffusion model for 1024 x 1024 using single 12GB memory GPU. However, as it is possible to train our model without showing the full-size image during training, we believe that it is extremely helpful for many studies on overcoming similar issues.
>
> **6. Typo: on P6, $T^f$ and $T^s$ should be $T_f$ and $T_s$ ; in Figure 3, diffsion --> diffusion**
>
> Thanks,	 We corrected the typos.
> ___
>
> **References**
>
> [1] Song et al, “Denoising Diffusion Implicit Models”, ICLR 2021

---

### Author Response · Authors · 2022-11-16
**General comment by Paper 5679 Authors**

We would like to thank the reviewers for their constructive and thorough reviews. We are encouraged that the reviewers think that our paper “verified the effectiveness of the proposed training method” (HWig) and “suggested an interesting idea of applying positional encoding for training a single score model for multiple scale of images” (VnnC, REZk, 4NKS).
We have updated our draft to further clarify the theoretical aspects along with typo corrections and additional experiments. In the following, we summarize the major changes that were made to the manuscript

**1. Presentation**

There were some typos on algorithms and texts, so we have corrected the typos. Also, some implementation details were missing and confusing, so we have clarified the details.

**2. Experiments**

There were some issues on the comparison method (SR3) on super-resolution task. We repeatedly tried to train new up-sampling diffusion models suggested in https://github.com/openai/improved-diffusion and https://github.com/Janspiry/Image-Super-Resolution-via-Iterative-Refinement. Again, we faced the same problem of __image noise and hue deviation problems__ for both implementations. Also, there are many similar cases which suffered from the same issue which can be found in the issues tab in the github repository. The only difference of the training was the batch size, so we guess there is a problem when training up-sampling diffusion model with low batch size (ours: 2, original: 128~256).
Therefore, for a fair comparison to improve the clarity of the experiment, we have added super-resolution result on AFHQ dataset via MCG [1]. Figure and table in the paper are updated.
Additionally, we have added results on some ablation studies in supplementary materials to justify the benefit of using positional encodings.
___
**References**

[1] Chung et al, “Improving Diffusion Models for Inverse Problems using Manifold Constraints”, NeurIPS 2022

---

### Decision · Program_Chairs · 2023-01-20

**Decision:**

Reject

**Justification For Why Not Higher Score:**

While the proposed idea is interesting and novel, almost all the reviewers have raised serious concerns regarding the experimental comparisons and reported results for prior works. Given these concerns, this submission does not appear to be ready for publication at ICLR.

**Justification For Why Not Lower Score:**

N/A

**Metareview: Summary, Strengths And Weaknesses:**

This paper introduces pyramidal denoising diffusion models that share one score function for different image resolutions. In addition to noise-injected images, the score function is given positional embeddings per pixel, allowing the model to adapt itself across different scales.

Pros:
- The core idea of incorporating positional encoding into the diffusion model for multi-resolution training is novel and interesting.

Cons:
- Most re-implemented baselines perform poorly compared to their official published results. Several reviewers have raised questions on whether experimental comparisons are sound.
- Sampling time comparisons are not performed in a controlled apple-to-apple fashion.
- Missing details and confusing descriptions at times

**Summary Of Ac-Reviewer Meeting:**

N/A